# Contribution of Maintenance 4.0 in Sustainable Development with an Industrial Case Study

**Yousra El kihel [1],\*, Ali El kihel [2] and El Mahdi Bouyahrouzi [3]**

1    IUT of Bordeaux-Bastide, Department MLT, University of Bordeaux France, CEDEX, 33076 Bordeaux, France
2    Faculty of Science and Technology of Settat, University of Hassan I Casablanca, Settat 26000, Morocco
3    Industrial Engineering Laboratory, National School of Applied Sciences, Mohammed First University, Oujda 6000, Morocco
\*    Correspondence: yousra.el-kihel@u-bordeaux.fr; Tel.: +33-0666969023

**Abstract:** Digitalization and digitization are topics for researchers and manufacturers. Integrating new technologies facilitates the collection of data from a company in real-time and processing them afterwards. In this context, the design and implementation of Maintenance 4.0 have become popular in the literature. Its objective is to minimize downtime, optimize energy consumption, and increase availability, utilization rate, and useful life of machines while ensuring environmental preservation and safety of personnel. Our contribution consists of setting up a specific digitalization methodology for companies wishing to switch to Maintenance 4.0 in order to contribute to sustainable development. The information obtained will be processed to carry out effective interventions to increase the reliability and availability of equipment. A case study of an industrial company was carried out where we implemented this methodology. As a result, we were able to increase the reliability of the machines, which has an impact on the environment by reducing energy consumption and the quantity of plastic waste. On the economic level, this led to an improvement in the Overall Equipment Effectiveness (OEE) and a reduction in product prices. Thanks to these technologies of digitizing maintenance documents (procedures, machine history, risk prevention) and the quick localization of machine failures, the hard work and risks are reduced.

**Keywords:** Industry 4.0; maintenance 4.0; sustainability; environment; risk; economy

## 1. Introduction

The concept of Industry 4.0 first appeared in Hanover in 2012; it represents a new generation of the industry of the future, and new technologies such as artificial intelligence, Internet of Things, and Big Data represent the basis of this industry [1,2]. Industry 4.0 is considered a step of digitalization and digitization of the manufacturing sector, it is based on four main bases: the huge increase in data, computing power and connectivity, new low-power wide area networks, and finally the emergence of analytical capabilities and artificial intelligence [3,4]. It is necessary to note that physical systems in Industry 4.0 are controlled by digital systems such as digital twins and Power BI (Business Intelligence) [5], decisions are made in an intelligent manner by the combination of the human part and the infrastructure part [6,7].

The technological and digital revolution is a major factor for industrial development and especially for developing the industry of the future. This development makes industrial operations more and more automated and technologically very complex [8]. This complexity can produce risks for the company and customers and damage the environment and the safety of people. Therefore, bad management of digitalization can negatively impact the production chain and consequently decrease the quality of the product and reduce the competitiveness of the company. For this reason, the implementation of technological solutions for the contribution to sustainable development has attracted increased

interest in recent years. Sustainability in the industrial sector implies that production lines operate with maximum efficiency, without wasting energy and without harming the environment or increasing the carbon footprint and waste. According to D. Brochard [9], sustainable development is managing human activity to meet the needs of the present without compromising the ability of future generations. It is concerned with three main areas: environmental, economic, and social. Thus, it is a way of organizing the company to consider both present and future imperatives, such as the preservation of the environment and natural and human resources or industrial risk and safety. As described by C. Herrmann, and S. Kara [10], sustainable manufacturing seeks to ensure that production will be carried out economically, considering the use of sources and ensuring social standards.

In this context, new digital solutions and technologies based on Industry 4.0 allow today to manage efficiently critical resources and energy, focusing on sustainability and safety of equipment and the environment. Among these solutions, a new maintenance paradigm, methods, and innovative tools have been developed. Maintenance must progress towards the requirements of Industry 4.0 to become in the same level with the industry today and meet the challenges of competitiveness [11]. In the context of Industry 4.0, this maintenance is often called Maintenance 4.0 or Predictive Maintenance. It is defined as a new maintenance paradigm that is based on three main components [12]:

- Real-time monitoring of the production line to detect all kinds of waste and failure.
- Diagnosis is the location and identification of the causes of anomalies under the failures observed.
- The prognosis is to have an estimation on the time of operation and on the evolution of the problem degradation to take good choices.

On the other hand, Maintenance 4.0 plays a critical role in the future industry for it represents a practical solution not only to monitor and diagnose the condition of a plant's facilities but also a solution to anticipate failures, identify early signs of failure, and plan maintenance activities [13]. Moreover, it is a new generation of maintenance that many sectors have adopted, especially those where reliability, safety, availability, efficiency, and quality, as well as environmental protection, are paramount [14–16].

Both authors, Haarman [17] and Mobley [18], stated that the impact of maintenance represents a total of 20% to 50% of total operating costs, and Maintenance 4.0 can increase the availability of machinery and plant performance, increase profits, and reduce the costs of spare parts inventory which leads to reduce maintenance costs. According to the authors of [19], Maintenance 4.0 is based on using historical data to detect trends in equipment behavior to predict the time of failure. Once the failure trends are identified and the failure time is predicted, the maintenance tasks will be planned. Several review papers [20,21] have also discussed suggestions, challenges, and future direction for Maintenance 4.0 on how to implement algorithms that prognosticate faults.

Effective Maintenance 4.0 offers technical, economic, and social benefits. It increases the availability of industrial systems on the one hand and extends their life cycle on the other hand [8]. From an economic point of view, it reduces the cost of repairs, which increases the company's profit. Maintenance 4.0 promises improvements in productivity, flexibility, and integration of production systems [22].

To respond to this digital transformation, companies must progress by transforming their processes through the development of various technological levers, which should promote decentralized decisions through system connectivity, digital transformation, and real-time communication while contributing to sustainable development.

To remedy these constraints, the main objective of our research work is to propose and develop a Maintenance 4.0 strategy while respecting the sustainable development axes.

The paper contains four main sections. The first section presents a bibliographical study on Maintenance 4.0 regarding the environmental, safety and energy efficiency challenges of sustainable development, and the state of the art has been developed in the same section. The second section deals with the methodology developed to successfully make the digital transformation from classical maintenance to Maintenance 4.0 while respecting

the notions of sustainable development. The third section presents the application of the proposed method in a confirmed case of an agri-food company that has chosen two strategic energy production units to digitalize, as well as a performance study in terms of sustainable development after implementing Maintenance 4.0. The results obtained are presented in the fourth section, and they confirm that Maintenance 4.0 increases the reliability of the equipment, which has a positive impact on the environment by reducing the consumption of energy and plastic waste, and on the economy by improving the quality and the cost of the products and on the safety, a clear improvement of the accidents and hard work.

## 2. Bibliographic Study

The implementation of Maintenance 4.0 solutions for the contribution to sustainable development has received increased interest in recent years. Thanks to technological, IT, and organizational advances. Maintenance 4.0 has become increasingly requested in the industrial sector, especially for companies that want to digitalize their processes and achieve a global plant performance. Over time, maintenance has evolved from Maintenance 1.0, based on a reactive and corrective approach, to Maintenance 4.0, which is focused on a predictive and anticipatory approach, passing through Maintenance 2.0 (a preventive approach) and Maintenance 3.0 (a proactive approach). The characteristics of each type of maintenance are presented in Table 1.

**Table 1.** Maintenance characteristics [23].

|  | **Maintenance 1.0** | **Maintenance 2.0** | **Maintenance 3.0** | **Maintenance 4.0** |
|---|---|---|---|---|
| Data source | Operator experience | Machine maintenance schedule | Operator, machine maintenance schedule, information systems | Multiple data sources (machine history, machine status) |
| Data collection | Manual collection | Manual collection | Semi-automated collection | Automated collection via sensors and IoT system |
| Data storage | Operator memory | Written documents | Databases | Cloud services |
| Data analysis | Arbitrary | Theory of reliability based on assumptions and intervals | Conventional algorithms | Artificial intelligence, especially artificial neural networks |
| Data transfer | Verbal communication | Written documents | Digital files | Digital files |
| Data management | Non-existent | Human operators | Information systems | Cloud, artificial intelligence, and data warehouse |

We then present a literature review to extract the number of publications on Maintenance 4.0. Therefore, the analysis is based on the different articles published (scientific publications, journals, project reports, or documents produced by consultants) on the Scopus platform from 2015 to 2021. As shown in Figure 1.

Figure 1 shows the number of publications related to the theme Maintenance 4.0 from 2015 to 2022, with a total of 6690 publications which shows the increase in the number of publications around Maintenance 4.0 in terms of volume of publications and the vital interest of the scientific and technical communities to move towards maintenance that is based on the technologies of Industry 4.0 to anticipate and diagnose defects and even to make decisions, all to increase the competitiveness and the companies competition.

Figure 2 shows the number of publications related to the Maintenance 4.0 theme and the sustainable development axes. According to our analysis of different articles, we found that different authors have emphasized the importance of Maintenance 4.0 to reduce the negative environmental impact in particular energy consumption [24]. Some authors have shown the Maintenance 4.0 contribution to reducing industrial risks and their impact on human health [10]. For other studies, the authors agreed on the contribution of Maintenance

4.0 on the reliability of machines and consequently the quality and cost of products and the productivity of companies [25].

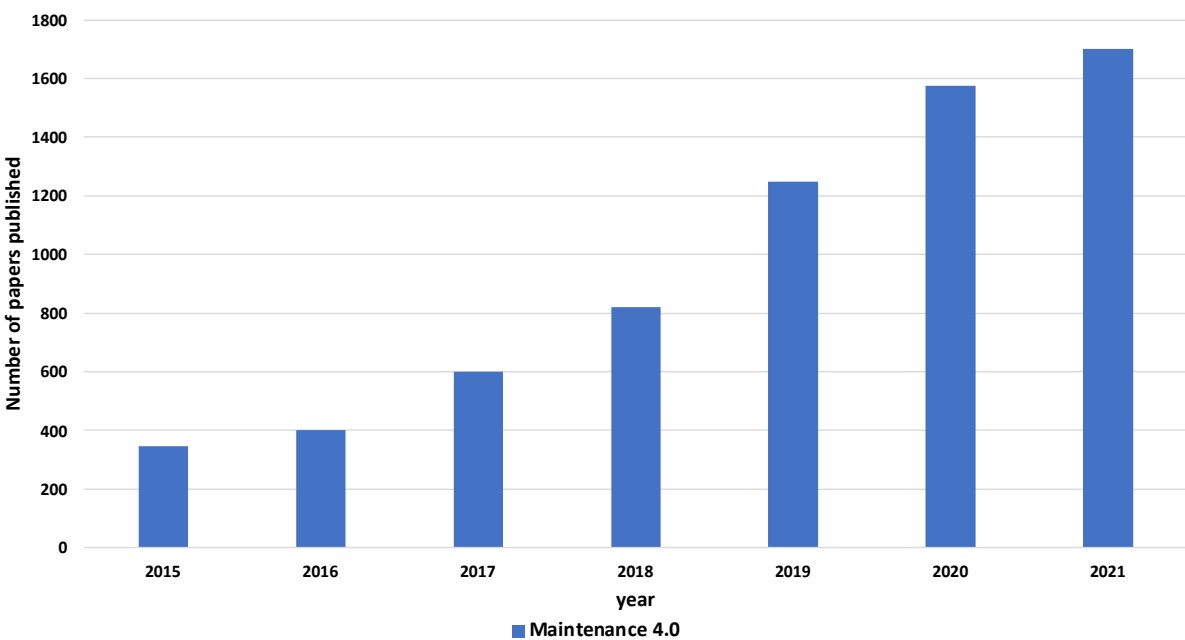

**Figure 1.** Number of Maintenance 4.0 related publication by year from 2015 to 2022 on Scopus database.

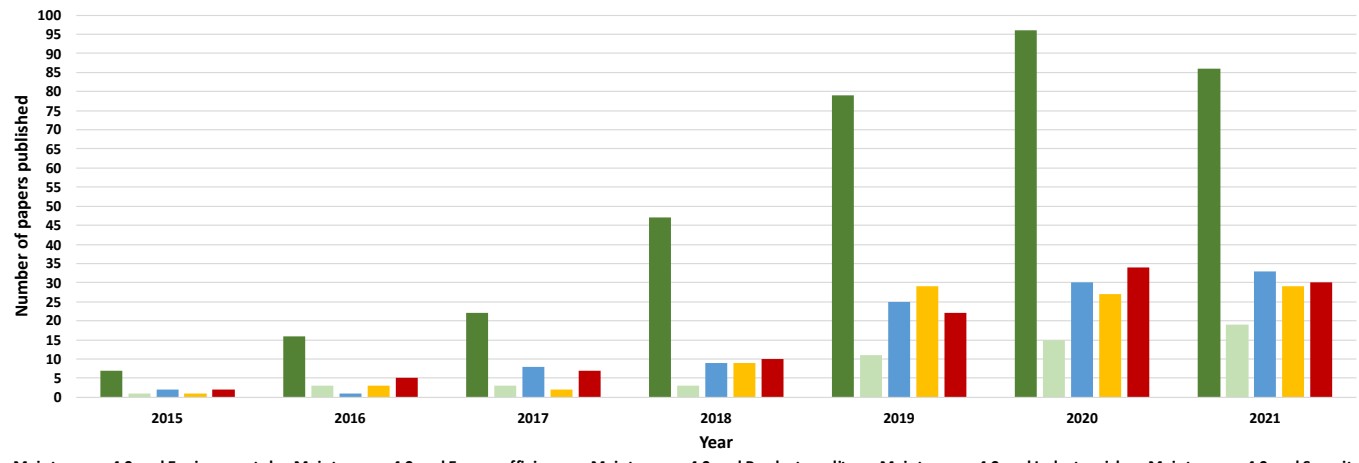

**Figure 2.** Number of papers published each year on Maintenance 4.0 technologies for sustainable development from 2015 to 2022.

There is a growth in the number of publications from 2019 onwards compared to previous years. In addition, we note the axis of the environment and the most important among the other axes of sustainable development.

From these two analyses, we note the intense interest of the scientific and industrial communities to achieve the shift to Maintenance 4.0 with respect to the axes of sustainable development. We also note that Maintenance 4.0 is more and more requested to solve the complexity of modern industrial systems, which constitutes a significant problem for the competitiveness of the companies at the national and international levels.

Another study was carried out, mentioned in Figure 3, to select the new technologies of Industry 4.0 qualifying as the most adequate for the Maintenance 4.0 and the sustainable development axes.

Figure 3 shows a variety of transformative technologies applied in Maintenance 4.0 that address sustainability. It can be noticed that there are many technologies in Industry

4.0 that address the problem of sustainability, but we see three technologies are the most used in sustainable development: artificial intelligence (AI), Internet of Things (IoT), Big Data. This result can be explained by the significant amount of data collected by IoT which requires more advanced processing by artificial intelligence.

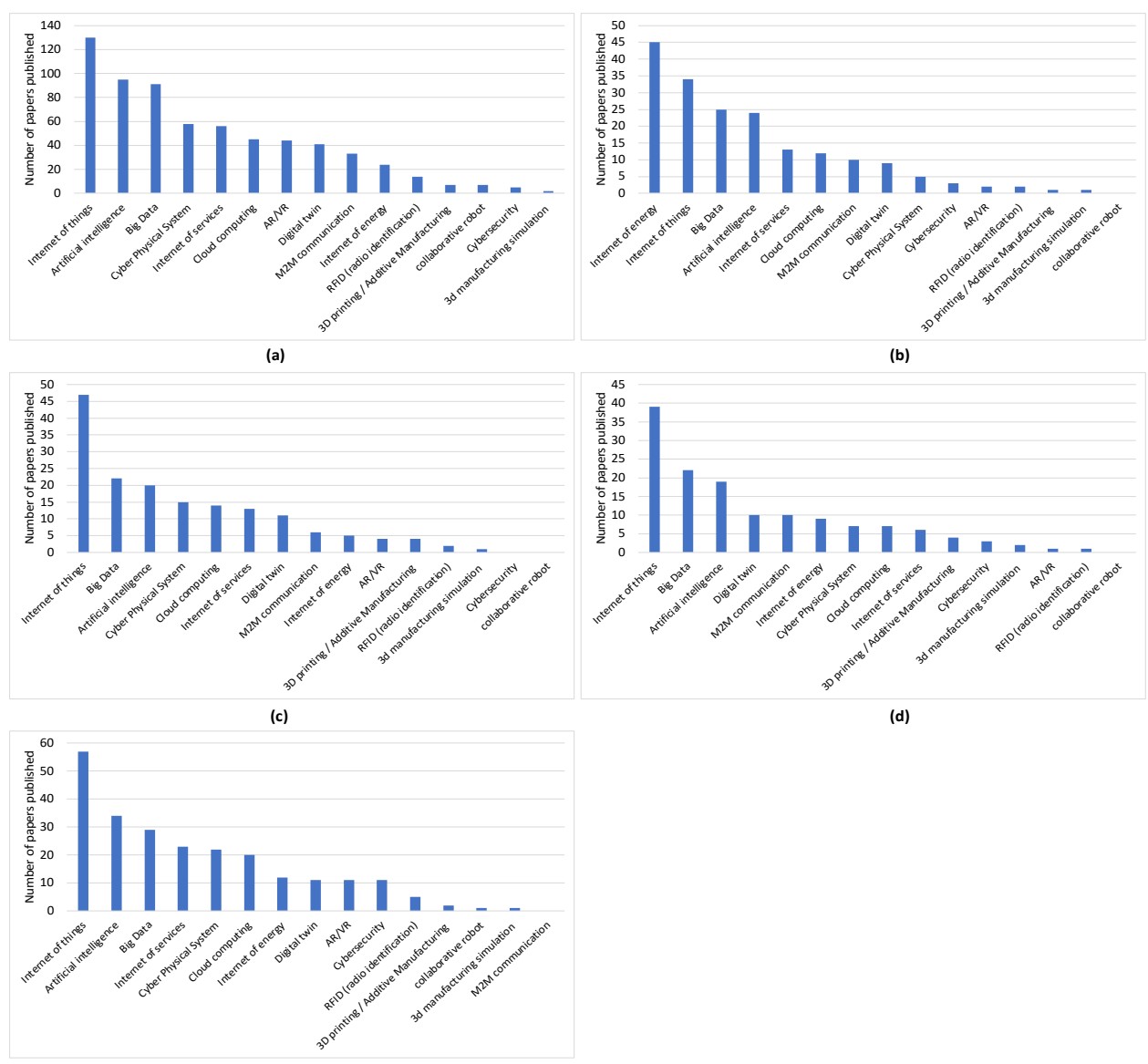

**Figure 3.** Statistics from the Scopus database on the number of publications: (**a**) Maintenance 4.0 and environmental; (**b**) Maintenance 4.0 and energy efficiency; (**c**) Maintenance 4.0 and product quality; (**d**) Maintenance 4.0 and industry risk; (**e**) Maintenance 4.0 and security.

Despite the growth of the number of publications on Maintenance 4.0 in the literature in recent years, only a few methods have dealt with the transition from classical maintenance to Maintenance 4.0. However, a large part of these methods targets diagnosis and prognosis. They do not deal with the digitization process from classical maintenance to Maintenance 4.0. We also note that few works have dealt with practical cases. In this regard, our original contribution of digitalization helps each company in a personalized way to design and pilot a transformation strategy towards Maintenance 4.0, prioritizing the axes of sustainable development to meet the legislative requirements, customers, and market competition. To validate this methodology, we carried out an industrial case study with very encouraging results.

### 2.1. The Contribution of the New Technologies in Maintenance 4.0

We also conducted a study to determine the relationship between these new technologies and the different functions of maintenance. Table 2 describes the contribution of new technologies in the different axes of Maintenance 4.0. We notice a strong link between the technologies and the different functions of maintenance.

**Table 2.** Contribution of the future technologies in maintenance 4.0.

| | IoT | Augmented Reality | Big Data | Cloud Computing | Digital Twin | AI | RFID | M2M | Additive Manufacturing | 3D Simulation | Cyber Security | Power BI |
|---|---|---|---|---|---|---|---|---|---|---|---|---|
| Diagnosis of the existing state | X | X | X | | X | X | | | | X | | X |
| Data acquisition | X | | | | | | X | X | | | X | |
| Data processing | | | X | | | X | | | | | X | |
| Monitoring | X | X | X | | | X | | | | | X | X |
| Diagnostic | X | X | X | | | X | | | | | X | X |
| Prognostic | X | X | X | | | X | | | | | X | X |
| Decision making | | | X | | | X | | | | X | X | X |
| Data visualization | | | | X | | | | | | | | |
| Sending data | X | | | X | | | | X | | | X | |
| Intervention | | X | | | X | | | | X | X | | |

### 2.2. Related Work

A review process of relevant publications was carried out following three steps: literature search, evaluation of the literature, identification of innovative approaches, and main ideas. In addition, this review process also captures the latest relevant reports developed by the Scopus platform in Industry 4.0 and sustainability, as shown in Table 3.

**Table 3.** The latest work on Maintenance 4.0 and sustainable development.

| No | Title | Main Ideas |
|---|---|---|
| 1 | Smart factory performance and Industry 4.0 (2020) [3] | - The benefits of predictive maintenance to reduce unplanned downtime and increase equipment life. <br> - The benefits and role of Industry 4.0 in sustainable development. |
| 2 | An overview of Industry 4.0 Applications for Advanced Maintenance Services (2022) [26] | - The authors of this article have classified maintenance services into three groups: basic services, intermediate services, and advanced services. <br> - Proposed an approach that helps to improve the supplier's competitiveness and achieve the customer's business objectives in terms of economy and sustainability. |
| 3 | Maintenance 4.0 Technologies for Sustainable Manufacturing—An Overview (2019) [25] | - The possibility of integrating IT technologies into the planning, monitoring, and analysis of maintenance processes in manufacturing companies. |
| 4 | Industry 4.0 for sustainable manufacturing: Opportunities at the product, process, and system levels (2021) [24] | - A comparative analysis examines the new technologies of Industry 4.0 and their potential impacts on sustainable development. |
| 5 | On sustainable predictive maintenance: Exploration of key barriers using an integrated approach (2021) [27] | - A list of Sustainable Predictive Maintenance (SPM) barriers has been developed to help industrial companies better understand the nature of these barriers in order to move forward with sustainability. |

**Table 3.** *Cont.*

| No | Title | Main Ideas |
|----|-------|-----------|
| 6 | Industry 4.0 applications for sustainable manufacturing: A systematic literature review and a roadmap to sustainable development (2022) [28] | - Identification of sustainability functions at the Industry 4.0 level and the value chain.<br>- Development of a structured model that is based on the axes of sustainability and compatible with Industry 4.0 concepts. |
| 7 | Sustainable Maintenance: a Periodic Preventive Maintenance Model with Sustainable Spare Parts Management (2017) [29] | - Provide an optimal maintenance model that takes into consideration conventional, environmental, and social costs. |
| 8 | Predictive Maintenance Planning for Industry 4.0 Using Machine Learning for Sustainable Manufacturing (2022) [22] | - An intelligent Maintenance planning 4.0 model has been addressed to increase manufacturing sustainability and reduce the reduction of breakdowns, failures, and material waste. |
| 9 | Industry 4.0 adoption and 10R advance manufacturing capabilities for sustainable development (2021) [30] | - Impact of Industry 4.0 adoption on manufacturing and sustainability capabilities.<br>- The implementation of Industry 4.0 technologies to drive companies to a higher level in sustainability and manufacturing capability. |
| 10 | Maintenance for Sustainability in the Industry 4.0 context: a Scoping Literature Review (2018) [31] | - A review of the literature in the field of sustainability and Industry 4.0 has been addressed by the authors.<br>- The authors highlight the relevance of the topic "maintenance and sustainability" and the importance of 4.0 enabling technologies in the industrial ecosystem. |
| 11 | Application of Modern Technologies for Planning Improvement and Saving on Costs in the Enterprise of the Industry 4.0 (2020) [32] | - Analyze and demonstrate the effectiveness of modern techniques to increase the company's profit while improving competition and equipment availability. |
| 12 | Integration of I4.0 technologies with maintenance processes: What are the effects on sustainable manufacturing (2020) [33] | - Identify the relationship between maintenance processes and Technologies 4.0.<br>- Analyze the positive effects of the integration of new technologies on the economic, environmental, and social dimensions of sustainable development. |
| 13 | Improving sustainability performance of heating facilities in a central boiler room by condition-based maintenance (2019) [34] | - A condition monitoring of a steam boiler by using condition monitoring and probable troubleshooting techniques has been discussed in this paper.<br>- The technique used shows that condition monitoring and diagnosis can be an effective strategy for steam boiler maintenance. |
| 14 | Towards intelligent and sustainable production systems with a zero-defect manufacturing approach in an Industry4.0 context (2019) [35] | - The article treats smart and sustainable production through the combination and integration of online predictive maintenance and material and environmental quality control in all phases of the process.<br>- Such an approach enables production systems in the manufacturing industry to transform and evolve towards zero defect manufacturing. |
| 15 | The engineering machine-learning automation platform (Emap): A big-data-driven ai tool for contractors' sustainable management solutions for plant projects (2021) [36] | - Predict risk in business and support decision making using new technologies such as machine learning (ML), cloud, predictive maintenance.<br>- Each module of the proposed method has been validated by case studies to ensure its performance. |

## 3. Proposed Methodology

Technological innovations and changes, customer expectations, and delivery times have reinforced the need for increased productivity. To this end, companies must now face fierce competition. They must progress by transforming their processes by developing various technological levers, promoting decentralized decisions through system connectivity, digital transformation, and real-time communication while contributing to sustainable development.

To meet the challenge faced in digital transformation, this research work aims to develop a new Maintenance 4.0 framework that contains all the monitoring, control, and optimization strategies at an enterprise level.

### 3.1. Proposed Architecture of Maintenance 4.0

Connecting machines, devices, and systems is essential for implementing effective, sustainable maintenance [36]. Therefore, it is necessary to design an infrastructure to collect the data of the monitored process and transmit them continuously and in real-time, as shown in Figure 4. As a result, our contribution is based on the design of an infrastructure based on the new technologies of Industry 4.0. This design proposes to transform conventional machines into connected machines by introducing sensors and electronic cards (data acquisition) on the one hand and transmitting the data from the latter for processing on the other hand.

**Figure 4.** The six steps of the integration of the proposed method.

The architecture illustrated in Figure 4 develops the ecosystem developed to bring up the information and the data of monitored systems by ensuring the acquisition in real time of the data and the dematerialization of the intervention on the industry and even the treatment and the anticipation of failures.

Figure 4 describes the six steps to follow to guide manufacturers in their project to integrate Maintenance 4.0 and even to transition to the future industry.

### 3.1.1. The Strategic Selection of Monitoring Systems

The company can choose all the units or only the most critical units and strategic installations to diagnose them, connect them, and implement Maintenance 4.0.

The first criterion to choose is the importance of the consequences of a failure and its repercussions. Some equipment or machines should not fail because the consequences are essential:

- Economic: with operating loss and delay penalties for suppliers, impact on product quality.
- Environmental: release of fumes, toxic gas, or liquid leaks that have a dangerous impact on the fauna and flora. Increase in energy consumption.
- Social: in case of risks following explosions, fires have severe consequences on the life of the people and the company.

### 3.1.2. Industrial Diagnosis to Identify Probable Failures Modes

To realize this diagnosis on the probable failures, we propose to realize this diagnosis by the FMEA tool (Failure Mode, Effects, and Criticality Analysis). On the other hand, we can exploit:

Feedback data: logical analysis of the failure/degradation, identification of causes, evaluation of consequences, etc.

The results of a questionnaire with the maintenance managers, the historical documents of the failed machine, etc.

This diagnosis is a critical element in the transition to Maintenance 4.0; it consists of a review or assessment conducted by experts to analyze the existing condition and allowed to:

- Examine the various failures of each particular facility.
- The selected strategic industrial facilities.
- Establish a detailed list of the distribution of all kinds of failures and their causes.
- Facilitate the choice of the Technologies 4.0 used.

### 3.1.3. Integration of Industry 4.0 Technologies

This diagnosis of Maintenance 4.0 refers to collecting and analyzing an industrial unit's available data to gather all the data and information related to the latter and facilitate the choice of new technologies. Therefore, the proposed diagnosis is based on two main steps to guide industrial companies to implement the new technologies of Maintenance 4.0. The first step is to establish a questionnaire which is a valuable tool in the diagnosis to know the existing state because it allows to take into account the reality of the company and its different departments, its strategy in digital transformation, and to know the degree of maturity of the maintenance of the equipment. The objective of this step is to know the degree of maturity of the company, taking into account the existing means. The second step is the setting of objectives and planning. For this phase, the company must set objectives compatible with its strategy to ensure the technical means in new technology and human resources by training skills and the appropriate time to implement Maintenance 4.0.

To realize an adequate diagnosis of Maintenance 4.0, it is preferable, if necessary, to call on external skills to provide expertise in areas.

### 3.1.4. Communication Protocol

In order to exchange data between the different components: sensor, acquisition card, and processing system, it is necessary to ensure that the exchange of information and data transmission is done in a comprehensible, secure manner with a control of error and data loss. To have the correct transmission, it is necessary to use communication protocols and data transmission modes such as Message Queuing Telemetry Transport (MQTT) as a standard messaging protocol for the Internet of Things and JavaScript Object Notation (JSON) as a standard format used to represent structured data and data frames similarly.

### 3.1.5. Data Processing

Algorithms based on artificial intelligence and, more precisely, neural networks can be introduced by exploiting the data for prediction. For the final part of this new design, Power BI (Business Intelligence)-based interfaces for data visualization were developed to dynamically present to the users the relevant content and results of the model in the form of curves, warnings, or visual notifications.

Regarding the decision-making, an algorithm favors the one that has a direct or indirect impact on the three axes of sustainable development: economy, environment, and risk.

### 3.1.6. Proposal of Integration Maintenance 4.0 Architecture

For facilitating the digital transformation, an architectural overview is presented in Figure 4, which illustrates our methodology for implementing sustainable Maintenance 4.0.

The various sensor data are collected and sent to a central system using a standard messaging protocol called MQTT [37], methods of processing the stored data, and, ultimately, dashboards that will serve as decision support tools, as shown in Figure 5.

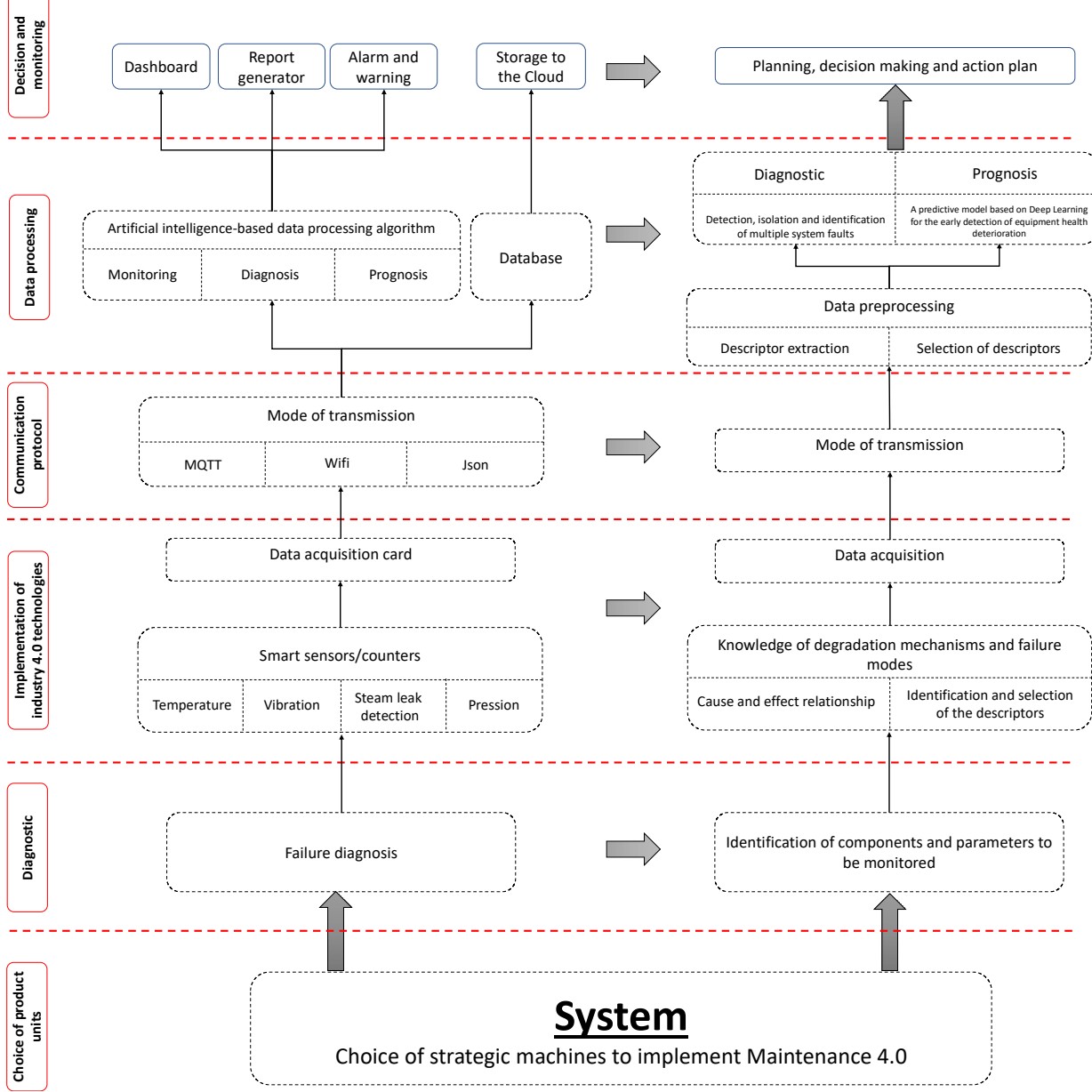

**Figure 5.** The six steps of integration the proposed method.

Finally, thanks to the connected equipment, it is possible to predict maintenance needs even before a problem occurs. This means optimized downtime and increased uptime.

### 3.2. Action Plan

Following the diagnosis and identification of probable failures, the industry 4.0 technologies, types of sensors, means of transmission, types of AI processing, and the approach that allows any company to move towards Maintenance 4.0 are determined. The next step is to study the feasibility of the project in particular:

A financial study and evaluation of the cost of the equipment and its implementation. The staff's training for data collection by the IoT, the transmission to the cloud, and the algorithms' processing know-how.

It is recommended for any company to choose efficient technologies, easy to implement with reasonable costs for its budget, and progressive implementation. This digital maintenance transformation positively impacts the three axes of sustainable development: economic, environmental, and risk.

## 4. Case Study

### 4.1. Company Presentation

In order to validate our research methodology on implementing Maintenance 4.0 in a company by integrating technologies 4.0, we chose an international beverage manufacturing company. The company SFAB agreed to collaborate with our laboratory by providing historical data on machines, workrooms, production, risks, and failures. The company produces beverages of 72 products with 12 flavors. The average daily production is 400,000 units and reaches 500,000 products per day in certain seasons (peaks).

Our work was to accompany this company to develop maintenance in two critical systems for its production process, whose failures have a negative impact on productivity and energy consumption. Therefore, Maintenance 4.0 was applied on two systems, which are:

- Steam and hot water production and distribution system represent a pressurized installation used to produce steam and heat water from the energy released by fuel oil combustion.
- Compressed air system, this system is used in all processes.

The company chose these two energy production facilities for several reasons, in particular to reduce energy consumption and avoid production stoppages due to breakdowns of these systems.

### 4.2. Thermal System

For the studied company, the thermal installation producing steam is critical for the different production processes.

The boilers constituting this installation are pressurized devices used to heat water and produce steam, generally through the energy released by the combustion of a fuel to feed the production facilities.

#### 4.2.1. Thermal System Modeling

In order to control and monitor the parameters in Maintenance 4.0, we modeled the thermal installation with its different components by three units: a steam production unit, a distribution unit, and a processing unit. For this purpose, we established an explanatory diagram that shows the implementation of our architecture on the thermal system illustrated in Section 4.2.4.

The system of production and distribution of steam and hot water consists of three central units:

- The equipment involved in the production of steam: For this part, we distinguish at the boiler's entrance the water treatment system and the burner using fuel oil, other elements such as the means of purging.

- The distribution circuit: This steam transport circuit comprises different organs, traps, valves, and piping.
- The processing circuit: The consumption concerns the production process, such as exchangers and production machines, as well as the control and regulation of temperature and pressure. The condensate recycling system plays the role of steam return.

4.2.2. Diagnosis of Maintenance Technologies Associated

The diagnosis consists of identifying the most probable failures from the history of each machine and questionnaire in order to associate Technologies 4.0 to the thermal installation.

After the diagnosis was made, several failures were detected. Among them, we can mention:

1.  Incomplete combustion at the burner

After analyzing the measured value of flue gas losses, it was found that these losses significantly influence the boiler's efficiency. The flue gas losses can be caused by an excessive air flow which can be due to:

- Incorrect adjustment of the burner.
- Maintenance problems such as poor air distribution or poor oil spraying.
- A clogged boiler: internal deposits (scale) and external deposits (soot) that limit the heat transfer between the boiler water and the flue gas.

The analyses of the flue gases showed that the temperature is very high, around 182 °C, which causes a decrease in efficiency. This temperature increase is due to the accumulation of soot and fly ash on the exchange surfaces. Therefore, it is necessary to integrate the sensors and clean these surfaces by performing soot removal, which must be carried out regularly within the predictive maintenance framework.

From the analysis carried out, we deduced the sensors necessary to detect the problem of incomplete combustion at the level of the burner. The following sensors are quoted below:

- Combustion analysis sensor, which allows precise and stable measurements and detection of whether combustion is complete or not.
- Flow meter sensor to control the oil flow and steam sensor and compare them with reference thresholds.
- Temperature sensor, which allows to measure the temperature of the combustion with high accuracy and transmit the data via a wireless system.

2.  Problems of the Losses of Circulation on the Surface

The degradation of the insulation of the pipe surfaces leads to energy losses by radiation. Several measurements were carried out at different times and at different boiler speeds by the thermographic instrument; the results are shown in Figure 6.

The losses by radiation are based on the principle that anybody in nature having internal energy diffuses heat by radiation according to the following law:

$$Q_{ray} = \varepsilon \cdot \sigma \cdot S \cdot \left( T_{corps}^4 - T_a^4 \right) \tag{1}$$

with:

$Q_{ray}$: Heat loss by radiation
$\varepsilon$: Thermal emissivity dependence on temperature
$\sigma$: Stefan–Boltzmann constant and its constant value which equals $5.67 \times 10^{-8}$ W/(m$^2 \cdot$K$^4$)
$S$: Surface radiation
$T_{corps}$ : Wall temperature
$T_a$: Ambient temperature

For more precision, several temperatures were taken as references at several wall points and then averaged. The calculation of the surface *S*, via Formula (1), of all the organs is estimated at 11.5 m$^2$ of a non-insulated surface with considerable thermal loss.

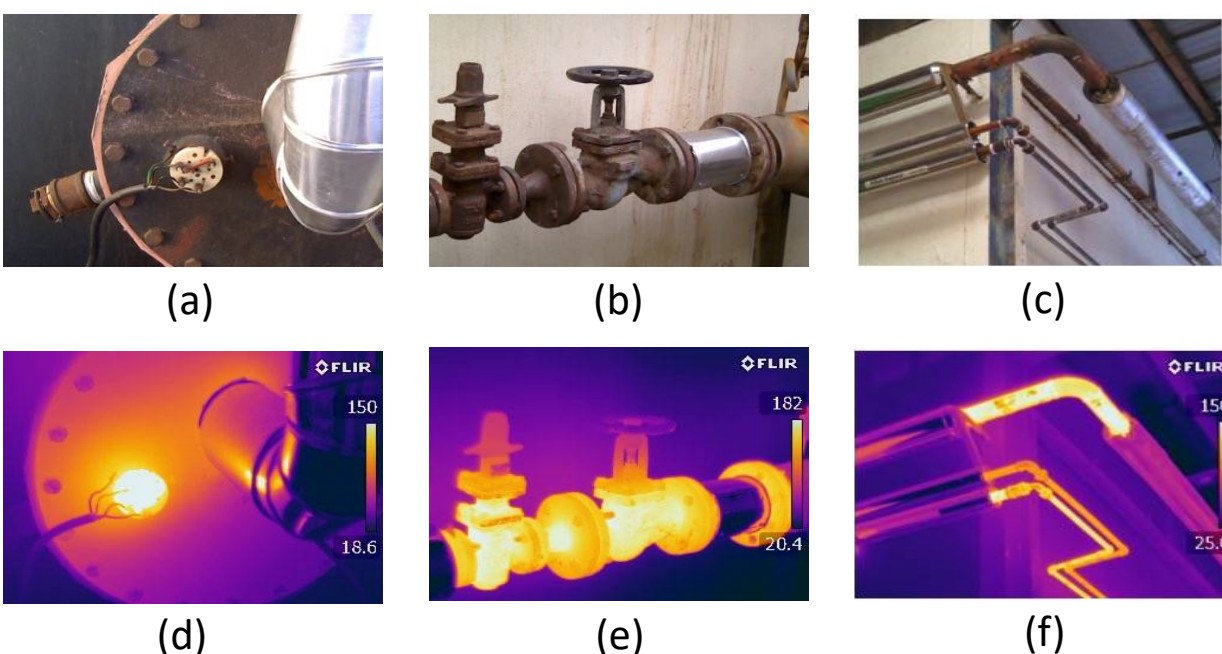

**Figure 6.** Infrared thermography-based diagnostics of the thermal installation: (**a**) Boiler thermal picture; (**b**) valve picture; (**c**) thermal heat pipes picture; (**d**) infrared thermal image of boiler; (**e**) infrared thermal image of valve; (**f**) infrared thermal image of steam transport tubes.

It was found that the number of leaks was significant at the valves, traps, and pipes. In order to detect these leaks of surface losses, which result from the degradation of the insulation of the surfaces, a thermal camera was introduced to capture the infrared radiation emitted by the bodies in the form of images and process them later.

3.　Leakage water pipes and valves

During the boiler's working, the water's transformation into steam leads to a high concentration of minerals in the water, which is the source of corrosion.

Corrosion causes boiler tube failure, scaling of heat transfer surfaces, and in turn, reduces performance and leads to under deposit corrosion, resulting in business interruption and costly maintenance. As a result, several Type 4.0 sensors were introduced to detect these leaks. These sensors include:

- Smart ultrasonic detector to detect leaks in traps and valves.
- Digital pressure gauge with analogue retransmission and a wireless transmitter to monitor pressure.
- Automatic sensor and controller for traps and valves to determine proper operation.
- All sensors are identified in the following paragraphs (see Table 5).

4.　Boiler efficiency drops

The energy released by the combustion reaction is not completely used in a boiler. There are indeed losses qualified as thermodynamic losses. This notion of losses allows the introduction of the notion of boiler efficiency, which the following formula can define:

$$\text{Efficiency} = \frac{\text{EnergyOutput}}{\text{EnergyInput}} \tag{2}$$

Two methods are widely used to evaluate the performance of a boiler, namely the direct and indirect methods.

In our case, we exploited the existing data provided by installed sensors to analyze and calculate the optimal point of the following parameters: efficiency ($R$), temperature ($T_c$),

and fuel flow rate ($\dot{m}_f$), as mentioned in Table 4, to achieve better efficiency. This calculation is obtained by an algorithm based on artificial intelligence that was developed [38,39].

**Table 4.** Sampling of the energy parameters database parameters which contains 111 data.

| $\dot{m}_f$ | $T_c$ | $R$ |
|---|---|---|
| 7.28 | 169.67 | 74.91 |
| 10.72 | 209.56 | 73.42 |
| 11.73 | 214.13 | 76.09 |
| . . . | . . . | . . . |
| 11.82 | 220.22 | 66.59 |
| 9.12 | 212.73 | 66.83 |
| 13.47 | 247.14 | 69.76 |

Table 5 summarizes the problems at the boiler.

**Table 5.** Energy installation problem.

| Unit | Organs | Failure Mode | Monitoring a Physical Phenomenon | Actions | Sensors |
|---|---|---|---|---|---|
| **Production** | Burner | Burner leak | Flow rate | Flow measurement | Fuel flow meter |
| | | Fuel flow rate | Pressure | Pressure measurement | Pressure sensor |
| | Water storage tank | Water leak Water circulates too fast | Flow rate | Flow measurement | Flowmeter |
| | | High temperature | Temperature variation | Temperature measurement | Temperature sensor |
| | Pump | Leakage | Leakage | Gas leak detection | Ultrasound |
| | | Temperature | Heating pump | Temperature measurement | Thermal camera |
| | | Cavitation | Vibration | Vibration analysis | Accelerometer |
| | Valve | Safety valve fail | System imbalance | Calibration | Imbalance sensor |
| **Distribution** | Piping | Pipe blockage Leak pipe Fouling effects of exhaust duct Without insulation | Flow rate | Flow measurement | Flowmeter |
| **Process** | Traps | Blocking Obstruction Wear and tear | Pressure | Pressure measurement | Pressure controller |
| | Piping | Leak pipes Without insulation | Flow/pressure | Flow and pressure measurement | Flow meter/manometer/ leakage sensor |
| | Cylinder | Mechanical failure Liquid leak | Vibration | Vibration measurement | Accelerometer |

### 4.2.3. Action Plan

After realizing the importance of dealing with the problems and their roots at the system level, an action plan was developed to implement the decisions in adequacy with the company's strategy, as shown in Table 6.

### 4.2.4. System Architecture Applied to Maintenance 4.0 on Steam and Hot Water Production and Distribution System

After the diagnosis, we applied the proposed architecture to the three units illustrated in Figure 7. This architecture aims to provide monitoring data to track the health status of each unit in real-time and to process this data through artificial intelligence-based algorithms. The rest of the steps are summarized in Figure 7.

**Table 6.** Proposed action plan for the boiler.

| Systems | Solutions |
|---|---|
| Energetic (Boiler) | <ul><li>Implementation of the different sensors to give a clear and timely view of the monitored system.</li><li>Deployment of an artificial intelligence-based model to calculate the maximum steam yield from minimum fuel flow and temperature [40].</li><li>Detect and measure energy losses.</li><li>Add a pressure regulator.</li><li>Check boiler settings.</li><li>Clean the air and fuel inlets as well as the exhaust duct.</li><li>Unclog the entire piping system.</li><li>Calculate surface losses.</li><li>Maintain the cleanliness of the heat exchange surfaces.</li><li>Recovery of fumes to preheat water by economizer/condenser.</li><li>Insulation of generators.</li><li>Definition of needs over time and optimization of supplies.</li><li>Optimized management of the boiler fleet.</li><li>Revision of the structure of the boiler room.</li></ul> |

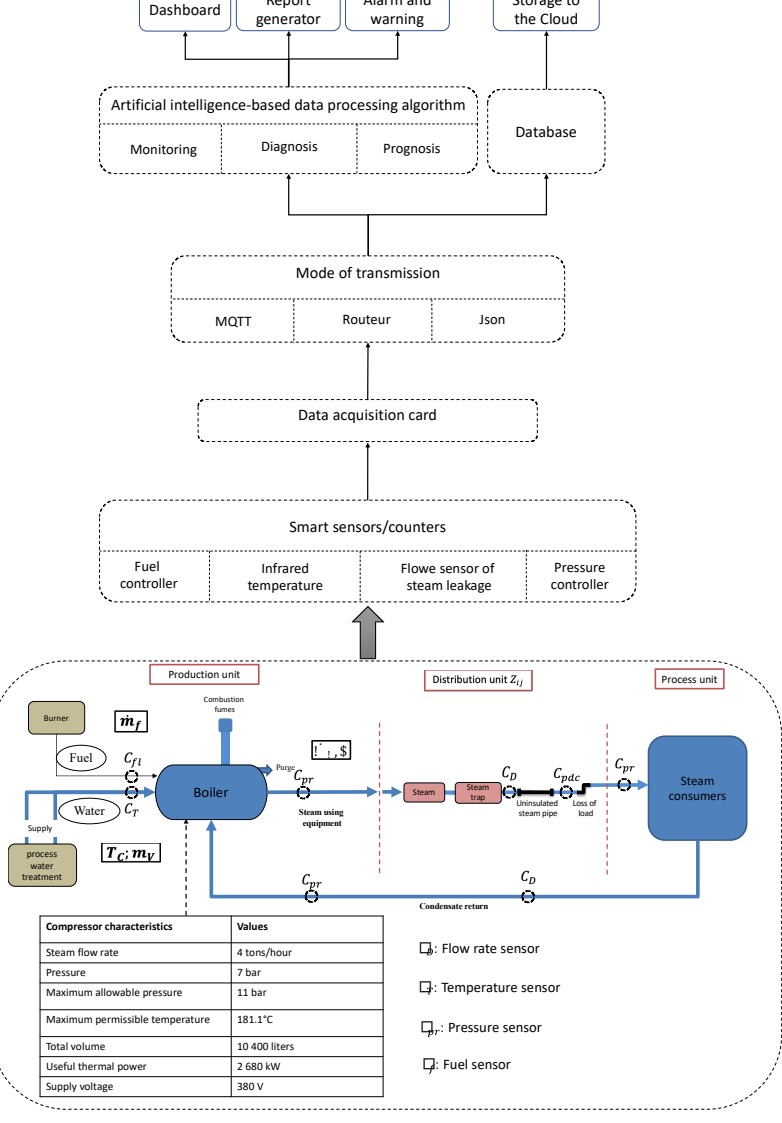

**Figure 7.** Implementation of Industry 4.0 Technologies and proposed architecture for the steam and hot water production and distribution system.

*4.3. Compressed Air System*

In industry, compressed air is a wholly adopted energy carrier that transports power, energy, and work. It is also a utility: its use can become very broad. In addition to being easily storable and transportable power carrier, it is also used as a drying, cooling, cleaning, waste ejection, transporting, and lifting.

4.3.1. Modeling of the Compressed Air System

In order to control and monitor the parameters in Maintenance 4.0, the entire compressed air production and distribution system was modeled, as shown in Figure 7.

The compressed air equipment include:

- Compressors
- Compressed air dryers
- Storage tank
- Filtration systems
- The distribution channel:
- Pipes sizes with length and diameter;
- The means of steam trap;
- The secondary expansion if it exists.

The consumers:

Compressed air consumers include all the production machines where pneumatic circuits are widely used for the transmission of movements: cylinders, etc.

This plant has been described using an explanatory diagram, mentioned in Figure 8, which contains: the production of compressed air, the different areas and organs constituting the distribution, and the process.

4.3.2. Diagnosis of Maintenance Technologies Associated

Diagnosing the compressed air installation allowed for measuring, monitoring, and analyzing the data. It also allows to:

- Evaluate the performance of the compressed air system;
- Evaluate the performance of the compressed air system—know the costs involved in the production of compressed air;
- Detect possible malfunctions;
- Correct the size of a new installation (change of equipment, capacity extension, etc.);
- Provide information to decision makers to support energy decisions.

Several modes of failure and losses in each unit were sampled; among these losses are:

1. Compressed air leak

Compressed air leaks have a very high percentage. In order to solve this problem, a very effective detection method by using an ultrasonic sensor of the type Technologies 4.0. This method can detect almost all system leaks, even those not heard during a production stop. A second method was used, based on the introduction of intelligent pressure and flow sensors to control, and verify the measured values with predefined thresholds of the compressed air system.

2. Problem of pressure drop increases

The pressure losses are due to the frictional forces of the airflow. The type of pressure drops distinguished: regular pressure drop. An intelligent pressure monitoring system is implemented along the compressed air circuit to detect pressure drops in the system, especially in the piping, to detect this type of problem related to pressure drop increases.

3. Problem of Air Compressor Vibration

In the standard case, the vibrations produced by the air compressor are very low. However, in some cases of compressor failure, the vibrations increase or exceed the norm due to the following reasons:

(1) insufficient supply of lubricating oil; (2) bearing failure; (3) position deviation of the wheel-center; (4) rotor air gap unqualified; (5) clogged oil filter; (6) shield anchor loose bolt. A computer-assisted vibration analyzer was introduced to analyze and interpret the compressor vibration signal for fault diagnosis. In addition, an infrared thermal sensor was implemented to measure the temperature of the compressor.

4. Increase humidity

Humidity is one of the most challenging factors in a plant, and even the most sophisticated equipment cannot control it perfectly. Humidity can lead to various problems, such as a decrease in the quality of produced air and the system's efficiency, which means that it is in the interest of the plant to keep the humidity level as low as possible.

Therefore, a system based on intelligent sensors and digital technologies was installed to control and regulate the humidity level quickly and automatically.

5. Filtering problem

Filters and valves are essential parts of a compressed air system, and their purpose is to filter the system's air to remove any impurities. Without this filter, dust, pollen, or some debris can pass through and infiltrate the system; it can cause a decrease in performance or an increase in consumption. Therefore, it is preferable to regularly check the condition of the filters to avoid any risk.

In our case, we installed a filter equipped with Technologies 4.0. We adapted to significantly reduce operating costs, ensure compliance with air purity standards, and extend the life of air networks, installations, and equipment.

6. Implementation of a Variable speed drive

Continuous operation of the compressors leads to overheating. To reduce air compressor energy consumption, it is necessary to control their speed. Proportional speed control can be provided by a digital variable speed drive that is sensitive to the variation in pressure-demand by the process and quickly and automatically provides the required pressure according to the demand and leading to an improvement in the energy performance of the plant.

This implementation of variable speed drive has several advantages:

- Adapting the pressure to the demand.
- Reduce energy consumption.
- Eliminate on/off sequences.
- Protect the compressor and increase its longevity.

A summary of compressed air system diagnosis shows several failure modes (see Table 7), which cause a decrease in plant efficiency and performance. As a result, switching to Maintenance 4.0, whose sensors are identified in Table 7, it is highly recommended to improve these energy performance indicators.

### 4.3.3. Action Plan

After realizing the importance of dealing with the problems and their roots at the system level, an action plan (see Table 8) was developed to implement the decisions in line with the company's strategy.

### 4.3.4. Proposed Architecture to Integrate Maintenance 4.0 on the Compressed Air Production System

The Industry 4.0 technologies were implemented in Figure 8 in the compressed air production system to move from corrective maintenance to Maintenance 4.0. Figure 8 shows the final architecture.

Following the proposed integration architecture on the two such systems, we move to measure performance indicators.

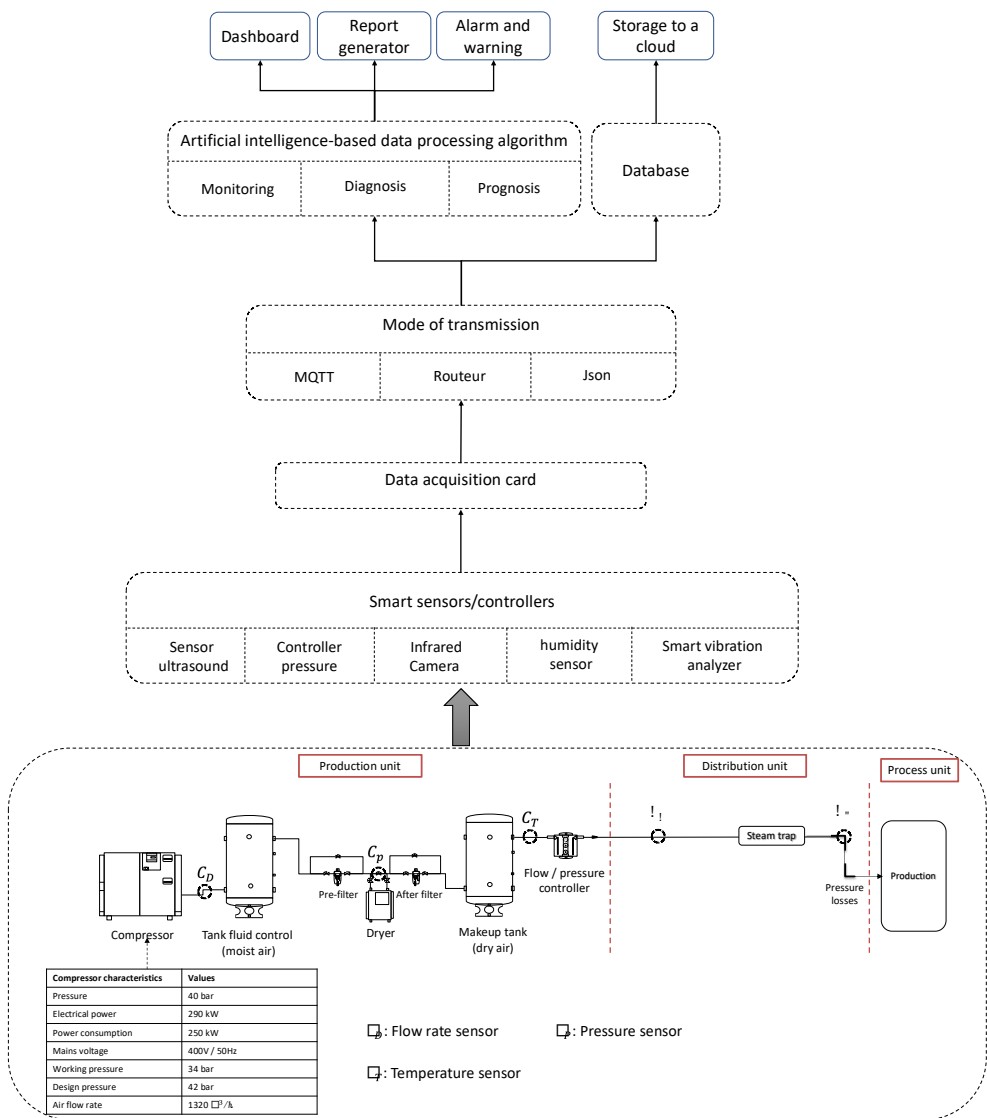

**Figure 8.** Implementation of Technologies 4.0 and the proposed architecture of the compressed air production system.

**Table 7.** Hot water production and distribution system problem.

| Unit | Organs | Failure Mode | Monitoring a Physical Phenomenon | Actions | Sensors |
|------|--------|--------------|----------------------------------|---------|---------|
| **Production** | Compressor | A device emits bad Vibrations | Vibration | Vibration analysis | Accelerometer |
| | | Low pressure or flow | Pressure and flow | Pressure and flow measurement | Mass flow and pressure sensor and controller |
| | | Oil consumption increases Air compressor not working | Low oil level | Oil level | Oil pressure sensor |
| | Reservoir | Leakage | Pressure | Pressure measurement | Pressure gauge |
| | | Uncontrolled air level | Air leak | Air level | pressure sensor |
| **Distribution** | Piping | Pipe blockage Leak pipe Fouling effects of exhaust duct Without insulation | Flow rate | Flow measurement | Flowmeter |

**Table 7.** *Cont.*

| Unit | Organs | Failure Mode | Monitoring a Physical Phenomenon | Actions | Sensors |
|---|---|---|---|---|---|
| **Process** | Traps | Blocking Obstruction Wear and tear | Pressure | Pressure measurement | Pressure controller |
| | Piping | Leak pipes Without insulation | Flow/pressure | Flow and pressure measurement | Flow meter/manometer/ leakage sensor |
| | Cylinder | Mechanical failure Liquid leak | Vibration | Vibration measurement | Accelerometer |

**Table 8.** Proposed action plan.

| Systems | Solutions |
|---|---|
| Energy (Compressed air) | – After a study was conducted, a variable speed drive was introduced. <br> – Implemented and integrated a leakage analyzer to detect system leaks and a thermal camera to monitor both the system temperature and the ambient temperature. <br> – Reduce pressure losses through early detection. <br> – Introduction of a piping system directly installed on the compressor to recover heat and evacuate hot air. <br> – Ultrasonic leak detection. |

*4.4. Study the Company's Performance in Sustainable Development after Implementing Maintenance 4.0*

4.4.1. Environmental Performance

1. Energy efficiency

After defining an architecture and implementing Maintenance 4.0 for both the compressed air and thermal energy systems, we opted for the energy management system to monitor the optimization of energy consumption. This performance measurement system, such as the ISO50001 v2018 standard, was applied to drive improvements and evaluate interventions. This standard allows to characterize the energy performance of the equipment, diagnose, and analyze the sources of failures and thus identify potential improvements. Figure 9 shows a diagram that summarizes the principle of the ISO 50001 standards. This system is considered a guide to achieving energy performance.

To calculate the company's energy efficiency, it is necessary to define the Energy Performance Indices (EPIs), which allow following the evolution of the global energy performance. The EPIs are specific to each company and are defined according to the company's strategic objectives. The ratios represent the leading performance indicators of the production unit. Three relevant indicators can give a clear idea of the evolution of energy consumption. The formula for each indicator is as follows:

$$\text{Ratio} - 1 = \frac{\text{Liter of drinks produced (L)}}{\text{Energy consumed (kWH)}} \tag{3}$$

$$\text{Ratio} - 2 = \frac{\text{Liter of drinks produced (L)}}{\text{Quantity of fuel consumed } (m^3)} \tag{4}$$

$$\text{Ratio} - 3 = \frac{\text{Liter of water consumed } (m^3)}{\text{Liter of drinks produced (L)}} \tag{5}$$

Our objective is to analyze these indicators to learn about the current state of the equipment to find the root causes of the increase in their energy consumption and consequently improve them by proposing adequate solutions in financial and technical terms. For this part of the study, the choice of indicators was of great importance, given its vital interest in coupling production and energy.

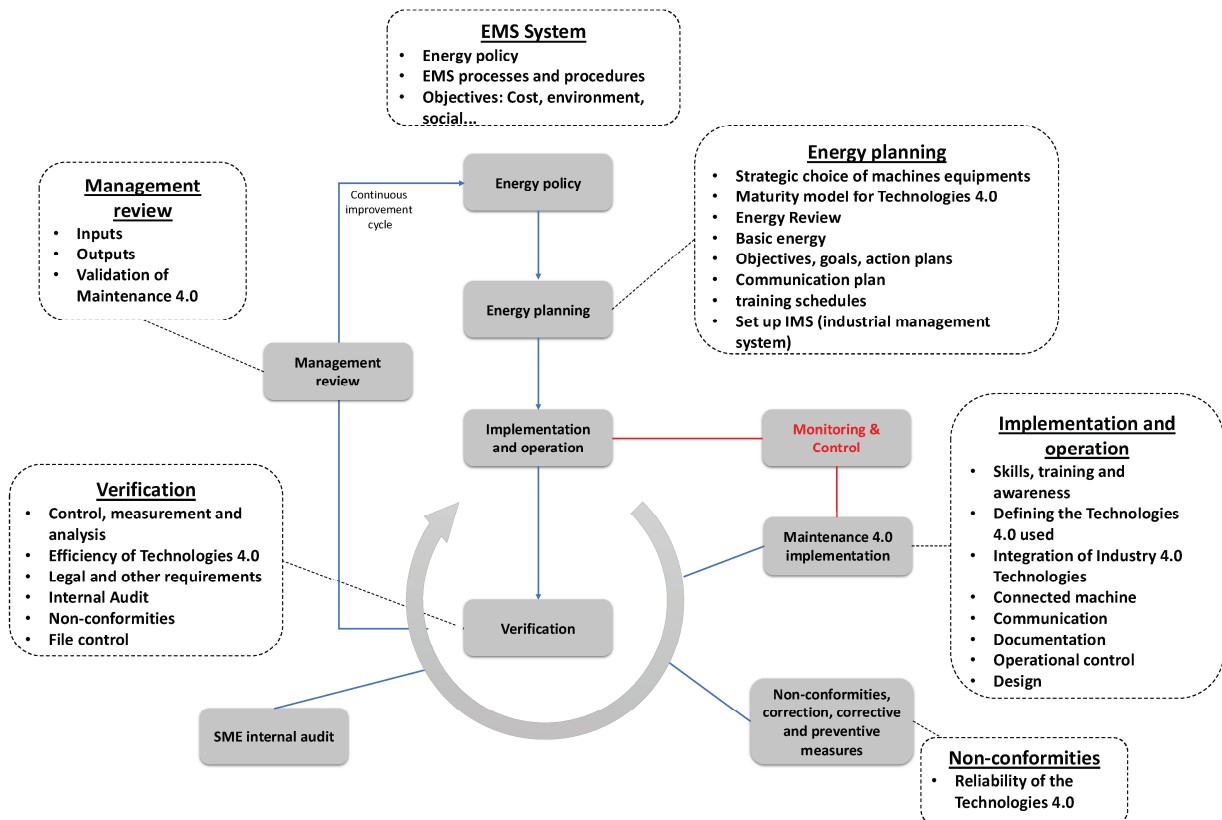

**Figure 9.** Iso 50001 energy management systems standard.

After studying and analyzing a database of recorded ratios in the company's information system, we obtained histograms of the ratios per year between 2016 and 2021. After integrating Industry 4.0 technologies and the proposed architecture, the ratio and the performance indicators increased remarkably from 2019. Figure 10 illustrates the improvement in system performance indicators as a function of the number of liter of beverages produced.

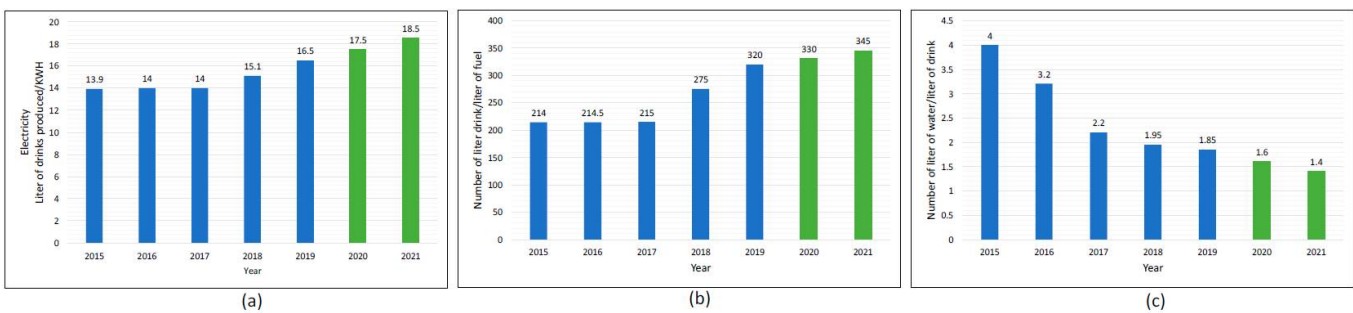

**Figure 10.** Performance indicator per liter of product by year: (**a**) Number of liter of beverage/kWh; (**b**) Number of liter of beverage/liter of fuel oil; (**c**) Amount of water/number of liter of beverage.

The improvement of these indicators (ratios) explains the effectiveness of the proposed methodology to implement Maintenance 4.0 in an industry whose objective is to optimize energy efficiency.

2.  Increase boiler efficiency

The boiler's efficiency plays a vital role in the production process. We used a database developed over the past years to analyze and define an optimum point between oil flow, combustion temperature, and efficiency. We found that the efficiency obtained varied between 65.34% and 76.97% (see Table 4).

To solve this problem, we developed a neural network algorithm to obtain the optimal oil flow and temperature point for maximum efficiency [39,40]. The optimal point is presented in Table 9.

**Table 9.** Good efficiency for the boiler system (obtained from the MSR).

| Fuel Flow Rate (kg/s) | Temperature (°C) | Efficiency (%) |
|:---:|:---:|:---:|
| 7.28 | 163.2375 | 76.9 |

We can conclude that it is necessary to improve energy efficiency because it is a source of growth, development, and excellence for the company. Monitoring energy efficiency is also an excellent way to quickly detect under-performance and implement action plans to improve the company's energy performance.

3.  Waste management

With the implementation of Maintenance 4.0, the number of failures was significantly reduced, which positively impacted the quality of the products, and consequently, the number of kg of plastic waste that results from these failures dropped a lot. We went from 680 kg of plastic waste to 98 kg. The consequences of the waste on the preservation of the environment are, in fact, immediate: a useless consumption of energy and raw materials that will require the manufacture of the rejected plastic part, which will be most often unusable.

In conclusion, the company motivated the staff by developing their skills by making them participate in fairs/conferences related to the environment and energy efficiency, in addition to investing in technology, which positively impacted the preservation of the environment within the company.

### 4.4.2. Economic Performance

1.  Indicator OEE

To measure the economic performance, we chose as an indicator the OEE (Overall Equipment Effectiveness), which represents a very effective performance indicator to evaluate the manufacturing productivity performance. This indicator reflects the percentage of manufacturing time that is productive. Measuring OEE and the underlying waste provides essential information on how to improve the company's manufacturing process systematically. OEE is the best indicator for identifying waste, evaluating progress, and improving the productivity of manufacturing equipment (i.e., eliminating waste). Our approach addresses all production losses: machine breakdowns, equipment defects, production slowdowns, changeovers, material shortages or breakage, and non-quality. Implementing Maintenance 4.0 of machines has led to better production quality. We express the OEE by the three rates of:

- Availability (TD) takes into account planned and unplanned downtime.
- Performance (TP) takes into account slow cycles and small stops.
- Quality (TQ) takes into account defects (including defective products).

The general formula of OEE is written as follows:

$$OEE = TD \times TQ \times TP \tag{6}$$

After the integration of Industry 4.0 technologies in industrial maintenance, the OEE was significantly improved, as mentioned in Figure 11; this is due to the real-time monitoring of all machine parameters: vibration temperature and leakage, which have increased the availability of machines by reducing breakdowns, product quality, and machine performance.

### 4.4.3. Risk Management Performance

1.  Operational risk management

There are many risks on an industrial site. To avoid these risks, guarantee employees' health and safety responsibilities, and protect the environment, Maintenance 4.0 responded

perfectly to this problem since it is possible to have information in real-time and anywhere and to interact between different systems.

For our case study, a diagnosis was made by analyzing the accident history of the past years, which shows that the most serious accidents occur during corrective maintenance operations, carried out after a breakdown, fatigue, and absenteeism from production.

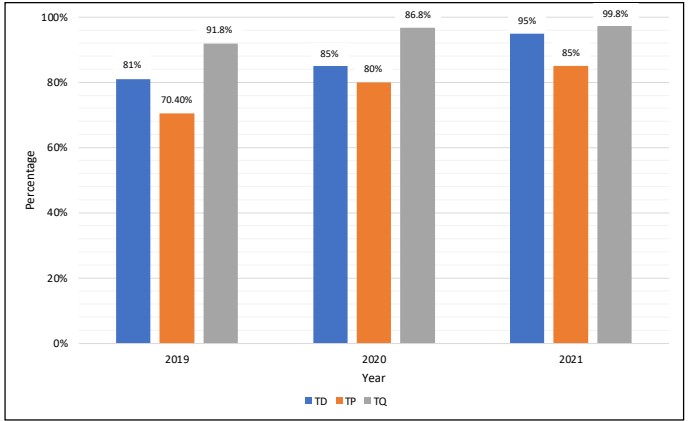
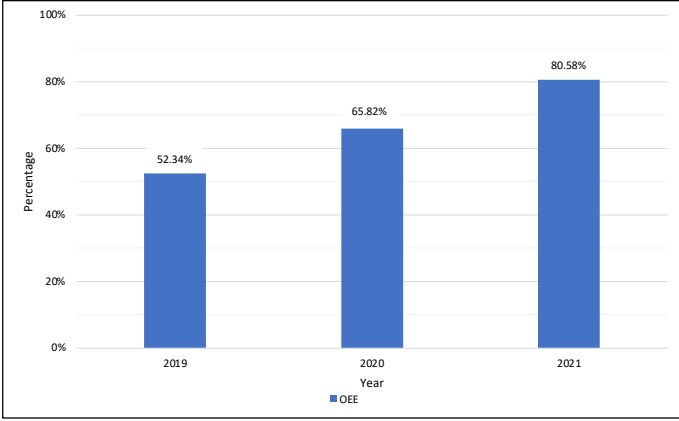

**Figure 11.** Evolution of OEE and its components per year.

We observed that maintenance operations are carried out in particular conditions and under a vacuum which can favor accidents and risks, and this is why these working conditions generally raise safety issues and require special attention to control the risks. We can mention the conditions identified in the two thermal and compressed air installations:

- Access to dangerous parts thermal, eclectic, pneumatic energies, etc.
- Urgency for interventions
- Stress (restarting as quickly as possible; the need for specialized knowledge and efficiency in variable and new situations)
- Increasingly complex equipment and technologies
- Dispersal of knowledge on equipment
- An increase in time constraints (affecting the quality of preparation and, therefore safety)
- An alteration in the transmission of information and organization (multiplication and coactivity of the participants)
- That of planning and procedures (defects in the maintainability of equipment or the management of the preparation of resources: materials, personnel, means, spare parts, documentation, methods)

2.  The contribution of Maintenance 4.0 in risk prevention

Maintenance 4.0 implemented in this company with connected equipment and digitalized documentation (machine documentation, procedures, drawings, etc.) allows for precise analysis of the data (physical parameters) recorded by the sensors connected to the machines. It becomes possible with the help of the GMAO, as a management tool to control the performance of the equipment in real-time in order to anticipate and predict breakdowns while minimizing operating costs. Consequently, it facilitates the control of industrial risks. In this context, industrial risk management is evolving and becoming more superficial, reliable, and predictive. In Table 10, we present the advantages and contributions of Technologies 4.0 to help maintenance operators intervene on equipment equipped with real-time information to avoid industrial risks.

Motivation and employee development by the number of formation and conferences on sustainable development, in particular, the risks prevention and the qualities of effective communication, aimed to help the business speakers understand the concept of danger and risk. Following this investment in human resources, it was noted in a reasonable time very satisfactory results:

- Number of major accidents is reduced to 100% and 95% for minor accidents.
- Absenteeism rate is reduced to 90%.
- Number of hours of intervention on the equipment is significantly reduced.

As a result of these changes, we noticed a climate of employee confidence, team spirit, motivation, and satisfaction.

**Table 10.** The contribution of Maintenance 4.0 in risk prevention.

| Technologies | Contribution on Occupational Safety and Health |
| --- | --- |
| Internet of Things Thermal, Vibration, Pressure, etc. | – Feed equipment data to the cloud by connecting industrial equipment online to learn more about their use. Integrated sensors can help anticipate any malfunction. <br> – The time factor is key to maintenance: detecting failures and a quick repair. <br> – The IoT alerts the operation managers through messages quickly sent on their user interface (computer, smartphone, tablet). <br> – Accurately monitor and analyze data recorded by sensors connected to machines. <br> – Monitor the performance of the equipment in real-time in order to anticipate and predict breakdowns. |
| Augmented reality | – Display precise information on the breakdown and locate it in real-time [32]. <br> – Visualize live data to see if acting on equipment or part is necessary. <br> – Connect the remote expert with a technician on the site of the breakdown. <br> – Quickly check the conformity of an assembly and the correct positioning of components. |
| Cloud computing | – It is the central tool for storing and harmonizing data: maintenance documentation, drawings, procedures, etc. <br> – Easy and quick access to data. |
| Artificial intelligence | – Use energy data to calculate the efficiency of the thermal installation. <br> – Advise users on the possible consequences of their decisions or actions to avoid risk. |
| Stacker crane | – An excellent tool to limit health and safety risks at work. <br> – The answer to arduous, dangerous, or low value-added work. It is mainly used in industry to handle parts in areas with dangerous heights |

## 5. Results and Discussion

Through this work, the company transformed classic preventive and corrective maintenance into more digitally evolved, efficient, and anticipatory maintenance. This transformation meets internal requirements regarding reactivity for adequate interventions on the two selected energy systems and market requirements. The work carried out is in the framework of sustainable development: economy, environment, and risk. Figure 12 summarizes the gains of the company resulting from this digital transformation.

For this company, several relevant economic factors are affected by Maintenance 4.0, which has improved the reliability and machine availability, performance rate, and the less non-quality product. The company achieved better production control and, consequently, a higher OEE. As a result, customer satisfaction and sales increased.

In terms of risk, with access to data concerning documentation (procedures, catalogs, machines, etc.) and the collection of maintenance parameter measurements (temperature, vibration, leaks, etc.), the company was able to anticipate stoppages and intervene rapidly, resulting in a reduction in the number of accidents.

For environmental sustainability, the energy management system of the ISO50001 standard was very judicious. The relevant indicators chosen and following several years showed a clear improvement in the energy consumption of water, fuel, and electricity. At the same time, this maintenance avoided purchasing many spare parts. This optimization had a positive impact on the reduction of carbon dioxide emissions. The reliability of the equipment contributed significantly to the reduction of plastic production waste (product packaging).

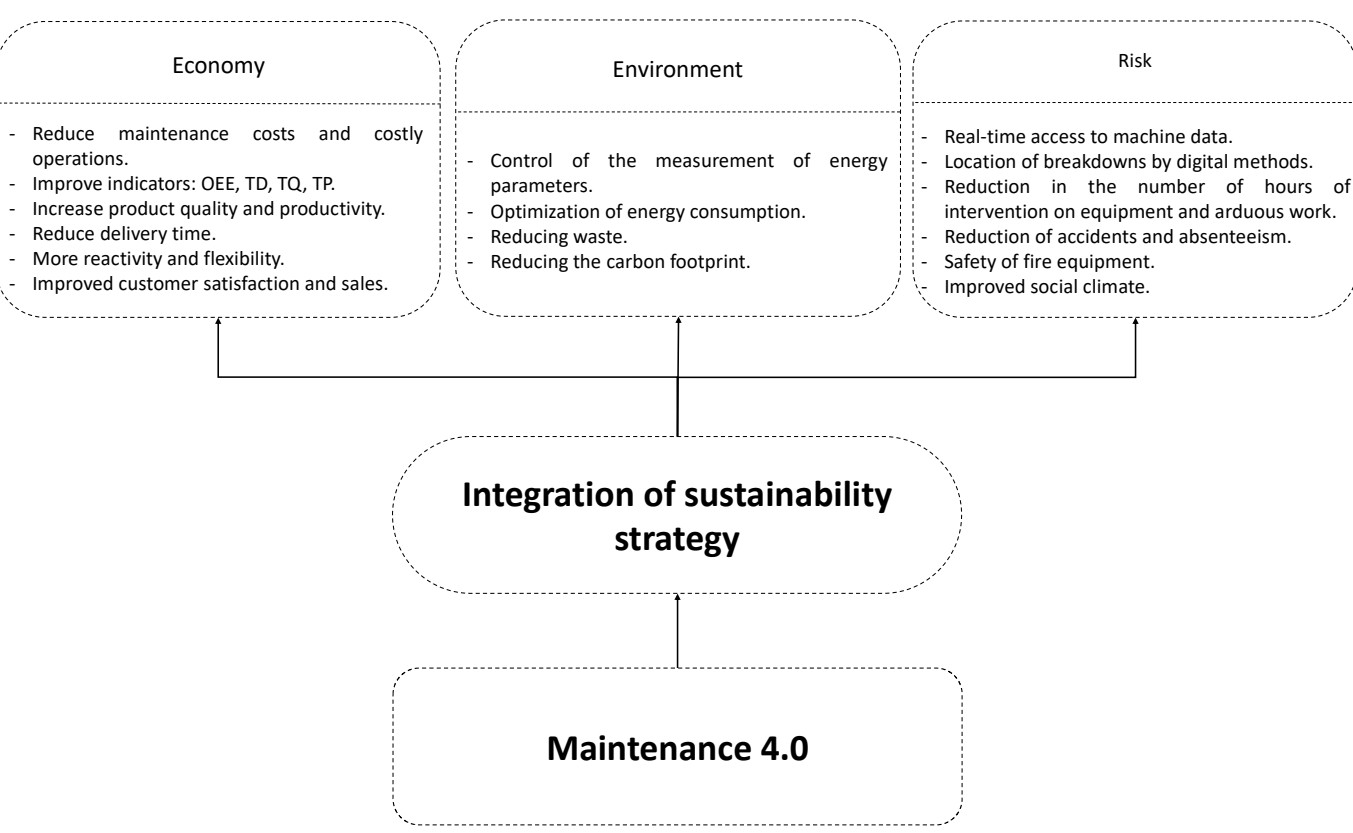

**Figure 12.** Impact of Maintenance 4.0 on sustainable development.

## 6. Conclusions

Currently, the search for a better sustainable performance, which balances a company's economic, risk, and environmental performance, is a significant concern for maintenance. Maintenance 4.0 is potentially an essential lever of action for the sustainable implementation of industrial production systems.

This paper has developed a methodology to establish a new Maintenance 4.0 framework that considers all the axes of sustainable development. To this end, the proposed method is divided into three main steps. This methodology is summarized in the choice of strategic equipment, a diagnosis to identify the likely failures followed by an assessment of the degree of maturity of the company and implement the associated 4.0 technologies and processing means to make decisions.

The application of this approach in a food industry company, where two energy installations are connected, has led to considerable performance:

- Environmental: improving energy performance indicators by reducing the consumption of fuel, electricity, and water consumption. A reduction in plastic waste results in increasingly reliable machines.
- Economic: The machine reliability and availability have improved the quality and cost of the products with an increase in productivity and OEE.
- Social: with these technologies, the company can access information from digitalized maintenance documents such as intervention procedures, machine histories, and rapid localization of failures. The increased reliability of the machines has reduced the number of hours of intervention on stopped machines and, consequently, an apparent reduction of accidents and hard work.

As a perspective, we recommend that the company continue to progressively integrate the technologies in all units of the company; investment in human resources must be made for the mastery of technologies and the independence of external experts. The

budget to succeed in this mission is a constraint for implementing such a methodology, but the progressive implementation allows a quick return on investment and encourages implementing others.

From the results of our work and benefits for the digital transformation of Maintenance 4.0, a Diagnostic 4.0 for the measurement of the degree of maturity must be developed, which has the role of identifying the sensitive components to monitor and optimize the number of sensors implemented.

**Author Contributions:** Conceptualization; methodology, software, formal analysis, investigation, Y.E.k.; data curation, writing—original draft preparation, Y.E.k.; A.E.k. and E.M.B.; writing—review and editing, A.E.k. and E.M.B. All authors have read and agreed to the published version of the manuscript.

**Funding:** This research received no external funding.

**Informed Consent Statement:** Not applicable.

**Data Availability Statement:** Data sharing not applicable.

**Conflicts of Interest:** The authors declare no conflict of interest.

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
