# Peer review of "Contribution of Maintenance 4.0 in Sustainable Development with an Industrial Case Study"

_sustainability, doi:10.3390/su141711090_

Round 1
Reviewer 1 Report
· This manuscript presents an interesting topic, Maintenance 4.0 in sustainable development with an industrial case study. Maintenance 4.0 and Sustainable Development are central issues in the manuscript.
· The concept of Maintenance 4.0 is lack of theory explanation. It must be explained in the Introduction section.
· The concept of Sustainable Development has not been explained at all.
· The relation between Maintenance 4.0 and Sustainable Development has also been clarified.
· The study conducted literature review describes the evolution from maintenance 1.0 to maintenance 4.0. However it is not clear what the characteristics of maintenance 1.0 to maintenance 4.0. How the classification of articles from 2015-2021 has not been explained related to maintenance 4.0.
· Figure 2 explained relationship between maintenance 4.0 and other concepts (claimed as sustainable development). However the academic justification has not been done for the relationships.
· Analysis and explanation of figure 1-3 are very lacking
· Section 2 and table 1-2 explained relationship between new technologies and the different functions of maintenance. However thera are no explaination about what is “new technology” and “functions of maintenance”
· Table 2 shows that there is confusion between the concept of industry 4.0 and maintenance 4.0 due to the unclear understanding of maintenance 4.0
· Proposed methodology (section 3) is not based on theory and previous research. The validation and verification process has not been carried out on the proposed methodology
· Existing case studies do not show strong linkages and explanations with the methodology.
Author Response
Hello,
Thank you for your pertinent remarks.
You will find attached the answers to your remarks
Sincerely

Reviewer 2 Report
Please see the attachment.

Author Response

(The authors gave the same response as above.)

Round 2
Reviewer 1 Report
The explanation of the characteristics of maintenance 4.0 has been presented but it needs to be further elaborated, especially related to case studies
Reviewer 2 Report
Accept in present form.